# Marine furanocembranoids-inspired macrocycles enabled by Pd-catalyzed unactivated C(sp$^3$)-H olefination mediated by donor/donor carbenes

Jiping Hao[1,5], Xueying Guo ⬤ [2,5], Shijun He[1,3,5], Zhongliang Xu[1,3], Lu Chen[1,3], Zhongyu Li[1], Bichao Song[1,3], Jianping Zuo[1,3], Zhenyang Lin ⬤ [2✉] & Weibo Yang ⬤ [1,3,4✉]

Biomimetic modularization and function-oriented synthesis of structurally diversified natural product-like macrocycles in a step-economical fashion is highly desirable. Inspired by marine furanocembranoids, herein, we synthesize diverse alkenes substituted furan-embedded macrolactams via a modular biomimetic assembly strategy. The success of this assembly is the development of crucial Pd-catalyzed carbene coupling between ene-yne-ketones as donor/donor carbene precursors and unactivated Csp$^3$–H bonds which represents a great challenge in organic synthesis. Notably, this method not only obviates the use of unstable, explosive, and toxic diazo compounds, but also can be amenable to allenyl ketones carbene precursors. DFT calculations demonstrate that a formal 1,4-Pd shift could be involved in the mechanism. Moreover, the collected furanocembranoids-like macrolactams show significant anti-inflammatory activities against TNF-α, IL-6, and IL-1β and the cytotoxicity is comparable to Dexamethasone.

[1] Chinese Academy of Sciences Key Laboratory of Receptor Research, Shanghai Institute of Materia Medica (SIMM), Chinese Academy of Sciences, Shanghai, China. [2] Department of Chemistry, The Hong Kong University of Science and Technology, Kowloon Hong Kong, China. [3] University of Chinese Academy of Sciences, Beijing, China. [4] School of Pharmaceutical Science and Technology, Hangzhou Institute for Advanced Study, University of Chinese Academy of Sciences, Hangzhou, China. [5] These authors contributed equally: Jiping Hao, Xueying Guo, Shijun He. ✉email: chzlin@ust.hk; yweibo@simm.ac.cn

Many natural macrocyclic small-molecules have evolved to interfere with protein–protein interactions, and often have been harnessed as probes for target validation and starting points for lead compounds for drugs discovery[1–6]. For example, marine cembranoids or furanobutenolide-based cembranoids which bear the alkene-substituted furan scaffold exhibit a wide range of biological activities, e.g., antitumoral, antimicrobial, and anti-inflammatory (Fig. 1a)[5,6]. Despite these valuable functions, gene expression limitations of soft corals and difficulty of resupply could hamper the sustainability of them. Therefore, the

development of strategies and methods to expeditiously access and enrich diverse natural furanocembranoids-like chemical space is highly desirable[7–12]. Inspired by natural products or privileged scaffold[13] and our interest in developing coupling reactions[14–17], we set out to create polysubstituted alkene furan-embedded macrolactams via a short and modular biomimetic strategy, which simply utilizes either the fundamental building blocks from living organism's endogenous ligands or mimics, such as amino acids or unnatural amino acids[18,19]. A retrosynthetic analysis indicated that a successive and concise Csp3-H carbene coupling, and

**Fig. 1 A modular strategy for the synthesis of natural furanocembranoids-like compounds via intermolecular Pd-catalyzed unactivated Csp3–H olefination mediated by donor/donor carbenes. a** Marine cembranoids or furanobutenolide-based cembranoids which bear the alkene-substituted furan scaffold. **b** Modular strategy for the synthesis of polysubstituted alkene furan-embedded macrolactams. **c** The design of Pd-catalyzed unactivated Csp3–H olefination mediated by donor/donor carbenes.

amidation could faithfully assemble these readily available building blocks, such as aryl bromides, natural, or unnatural amino acids, and enynones, into the target molecules (Fig. 1b).

Over the past decade, the Pd(0)-catalyzed cascade C(sp³)–H bond activation/formal 1,4-Pd shift process has emerged as a powerful method for organic chemists to construct complex molecules that might be difficult to synthesize by other methods. This kind of method relies on the catalytic generation of σ-alkylpalladium species and could be amenable to coupling with a wide range of organic trapping reagents. In this context, various efficient trapping transformations of σ-alkylpalladium have been developed after Dyker's pioneering work[20–22]. For example, Buchwald and co-workers disclosed C(sp³)–H arylation and amination via formal 1,4-Pd shift[23,24]. Very recently, based on a similar strategy, elegant Pd(0)-catalyzed cyclopropanation and amino- or alkoxycarbonylation were successfully realized from Baudoin's group[25,26]. Despite these significant achievements, the exploration of reactions to trap σ-alkylpalladium via C(sp³)–H bond activation/formal 1,4-Pd shift process is still highly appealing. In this regard, we questioned whether enynones as donor/donor carbene precursors can be utilized to trap σ-alkylpalladium via the aforementioned strategy. If successful, such a protocol would not only provide a convenient synthetic route for assembling our building blocks through Csp³-H carbene coupling but also can further expand the trapping reaction model. However, there are three challenging issues to be addressed in this scenario. First, Csp³-H carbene couplings have two distinct pathways: (I) metal carbene complex first formed, and subsequent Csp³-H bond activation/migratory insertion occurred. (II) Csp³-H bond activation initially occurred followed by carbene formation/migratory insertion. Compared to path II, the path I have been widely investigated[27–39]. In recent years, while efforts have also been devoted to the investigation of Csp³-H carbene couplings involving path II, they are limited to the examples on acceptor carbene cross-couplings of activated[40,41] or unactivated[42] Csp³-H bonds. To the best of our knowledge, transition-metal-catalyzed unactivated Csp³-H bonds functionalization/donor carbene insertion via path II have never been documented. The reason could be that the electrophilic capacity of donor/donor carbene is relatively weak. Second, metal-mediated cleavage of unactivated Csp³-H bonds is often significantly slower than metal-mediated cleavage of activated Csp³-H bonds. It has been recently demonstrated by Wang and co-workers who realized a donor/donor carbene cross-coupling of allylic Csp³-H substrates[43]. Finally, trapping of σ-alkylpalladium event involves metal carbenoid intermediates. Such intermediates are frequently plagued by competing for oxidation and dimerization[39].

Herein, we report the intermolecular Pd-catalyzed unactivated Csp³-H olefination mediated by choosing ene-yne-ketones and allenyl ketones as donor/donor carbene precursors via path II[32,44–50]. Accordingly, these reactions are distinguished by their high stereoselectivity and wide substrate scope including several drug derivatives. DFT mechanistic studies reveal that a formal 1,4-Pd shift could be involved[23,51–58]. The unique features of these alkene substituted furans are illustrated as building blocks for the construction of anti-inflammatory[59] macrocyclic targets.

## Results

**Reaction optimization.** Stimulated by the challenges of our synthetic target macrolactams, we first examined the module assembly by optimizing unactivated Csp³-H olefination. 1-Bromo-2-(tert-butyl)benzene and 3-(4,4-dimethylpent-2-yn-1-ylidene)pentane-2,4-dione were selected as the model substrates and a number of reaction parameters such as base, ligand, Pd

catalyst, and solvents were screened. After considerable experimentation, we were pleased to discover that a simple cocktail containing [PdCl(allyl)]₂ (5 mol%), ᵗBuXphos (30 mol%), and NaOAc in DMF at 100 °C in DMF established the reaction conditions, affording compound **3aa** in 76% isolated yield with high stereoselectivity after 4 h (Table 1, entry 1). This Csp³-H olefination is distinctive from Martin and co-workers' recent work[42], in which they described an interesting Pd-catalyzed [4 + 1] cycloaddition of diazo esters. A series of control experiments were also conducted to validate the role of each parameter. Not surprisingly, the examined parameters were all essential for this transformation. The use of either DIPEA or KOAc did not further improve the yield of the desired product **3aa** (Table 1, entries 2 and 3). Notably, the ligand appears to be important, as replacing ᵗBuXphos with Xphos or Brettphos provided **3aa** in a much lower yield and no reaction occurred in the absence of ligand (Table 1, entries 6, 7, and 8). In addition, a diminished yield was observed when Pd(MeCN)₄(OTf)₂ or Pd(OAc)₂ was employed (Table 1, entries 9 and 10). The influence of the solvents was also investigated. While similar efficiency was obtained using DMA, only traces of product were obtained in THF and no detectable amount of **3aa** could be found in acetonitrile (Table 1, entries 13 and 14).

**Substrate scope.** After determining the optimal reaction conditions, we turned our attention to evaluate the scope of this Pd-catalyzed intermolecular unactivated Csp³-H bond olefination with ene-yne-ketones as donor/donor carbene precursors. As shown in Fig. 2, our Csp³-H carbene olefination method turned out to be widely applicable regardless of the electronic variations at the *para* and *meta* positions on the aromatic ring of the aryl bromides (**3aa–3ea**). Likewise, the naphthyl bromide employed for the synthesis of **3fa** served well as a partner in the reaction. Gratifyingly, functional groups on the tertiary alkyls including cyano and ester are compatible (**3ga–3ha**), although aryl, secondary, and primary alkyls are not reactive probably due to steric hindrance or β-H elimination[60–63]. Particularly interesting was the observation that the aryl bromide substrate substituted with free amine did not interfere, providing **3ja** in a good yield without traces of the N–H bond carbene insertion product being observed. Remarkably, the ene-yne-ketones containing ketone, ester, and heterocyclic ring can be successfully transformed into corresponding products (**3ab–3ad**) in good to excellent yields (77–91%).

To evaluate the generality of the protocol, alternatively, we investigated this Csp³-H carbene olefination process using allenyl ketones as donor/donor carbene precursors[46]. As illustrated in Fig. 3, a wide range of allenyl ketones with electron-donating or -withdrawing substituents were well tolerated and a series of alkene derivatives substituted with dihydrofurans were obtained. Generally, reactions of allenyl ketones with electron-donating substituents attached to a phenyl ring proceeded in higher yields than those having electron-withdrawing groups (**5ab**, **5ac**, **5ad**, **5ah**). Moreover, the relative configuration of **5ab** was unambiguously assigned by the X-ray crystal structure analysis (Supplementary Data 1). Particularly, substrates bearing furanyl and thienyl functional groups were also amenable to the standard conditions, which provided the pharmaceutical bis-heterocyclic compounds in decent yields with excellent stereoselectivities.

The identification of lead compounds greatly benefits from fragment-based drug design and the ability to directly modify the privileged scaffolds. Therefore, to highlight the potential application of these Csp³-H bond carbene coupling reactions in medicinal chemistry, late-stage cyclization/olefination of complex and bioactive molecules was subjected to our established protocol.

**Table 1 Catalyst screening and optimization of the reaction conditions.**

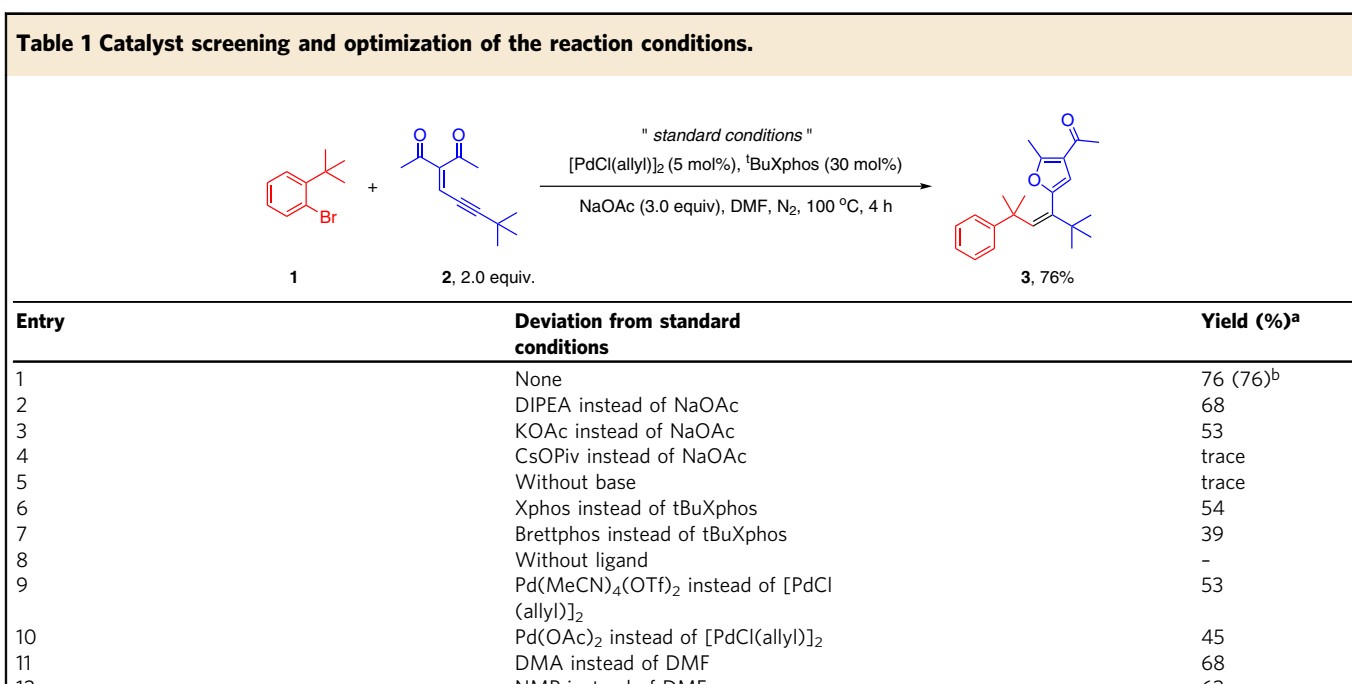

| Entry | Deviation from standard conditions | Yield (%)[a] |
|---|---|---|
| 1 | None | 76 (76)[b] |
| 2 | DIPEA instead of NaOAc | 68 |
| 3 | KOAc instead of NaOAc | 53 |
| 4 | CsOPiv instead of NaOAc | trace |
| 5 | Without base | trace |
| 6 | Xphos instead of tBuXphos | 54 |
| 7 | Brettphos instead of tBuXphos | 39 |
| 8 | Without ligand | – |
| 9 | Pd(MeCN)$_4$(OTf)$_2$ instead of [PdCl(allyl)]$_2$ | 53 |
| 10 | Pd(OAc)$_2$ instead of [PdCl(allyl)]$_2$ | 45 |
| 11 | DMA instead of DMF | 68 |
| 12 | NMP instead of DMF | 63 |
| 13 | THF instead of DMF | Trace |
| 14 | MeCN instead of DMF | – |

[a]Reaction conditions: **1a** (0.05 mmol), **2a** (0.01 mmol), catalyst (5 mol%), ligand (30 mol%), base (3.0 equiv), solvent (0.5 mL) at 100 °C for 4 h. [a] NMR yields were reported using CH$_2$Br$_2$ as the internal standard. [b] The yield of isolated.

Remarkably, the alkenes substituted with furans and dihydrofurans products derived from modified Repaglinide, Isoxepac, Mycophenolic acid, Adapalene, and Dehydrocholic acid were synthesized in moderate to excellent yields (Fig. 4). For example, Repaglinide, an antidiabetic drug used to control blood sugar in type 2 diabetes mellitus, had also been installed with 1-bromo-2-(tert-butyl)benzene and subjected into this protocol, gave access to the product **3ka** in an excellent 92% yield. Notably, starting from Isoxepac, a non-steroidal anti-inflammatory drug with analgesic activity, which was successfully converted to furan or dihydrofuran-containing Isoxepac (**3la**, **5lc**) in 84% and 74% yield, respectively.

**The construction and anti-inflammatory activities of poly-substituted alkene-embedded macrolactams.** Once the crucial connection of the aryl bromides and enynone building blocks was successfully established, we next selected different natural or unnatural amino acids and attempted to assemble them to the macrolactams via a short and modular biomimetic strategy. With 6–8 steps, eight polysubstituted alkene-embedded macrolactams (**6a–6h**) were efficiently assembled (Fig. 5). To explore whether these alkene-embedded macrolactams could successfully exhibit pharmacologically relevant features, the macrolactams **6a–6h** were investigated for the inhibitory effects on inflammatory mediators by lipopolysaccharide (LPS)-induced inflammatory responses in RAW 246.7 macrophages. The results showed that **6g** exhibited prominent inhibitory effects on the production of TNF-α, IL-6, and IL-1β with IC$_{50}$ values of 0.45, 1.59, and 0.59 μM, respectively (Fig. 6a). It should be noted that these pro-inflammatory cytokines are critically involved in the process of inflammation, immunity, cell survival and apoptosis, and metabolic diseases[64–66]. Both **6g** and **6h** were approximately ten times more potent in the inhibitory activity on IL-6 than the drug Dexamethasone, the widely used corticosteroid medication to

relieve inflammation (see the Supplementary Information). More importantly, they did not show obvious cytotoxicity at the indicated concentrations compared to Dexamethasone. We further examined the effects of **6g** on the activation of NF-κB signaling pathway induced by LPS in RAW246.7 cells. As expected, **6g** could abrogate the phosphorylation of IKK-α and degradation of IκB-α, an inhibitory protein of NF-κB nuclear translocation. Further, phosphorylation of NF-κB was also suppressed by **6g** in a concentration-dependent manner (Fig. 6b)[67]. The current preliminary pharmacological results indicated a promising prospect of **6g** to be developed as an anti-inflammatory agent, with competitive potency and safety advantage.

**Mechanistic studies.** Apart from the scope of these conversions and intriguing anti-inflammatory activities, we were also interested in the reaction mechanism. Two possible catalytic cycles are shown in Fig. 7. To gain insight into the proposed catalytic cycles and see which cycle is more favorable, we carried out DFT calculations to investigate the detailed mechanism (Supplementary Data 2). Although similar mechanistic pathways have been proposed by others[31], the DFT calculations of Pd(II) shift are still uncovered to date.

Figure 8 shows the energy profiles calculated. Considering the sizes of ligand and substrate, we start with the complex **A** in which the Pd (0) metal center coordinates with the ligand L and the substrate aryl bromide (Fig. 8a). The use of the simplified model ligand is for the purpose of theoretical simplicity. This simplification would not affect the conclusion we will make because our main objectives are to compare the two possible catalytic cycles. The errors introduced by the simplification is expected to be canceled out as a result of the comparison. For selected steps, we carried out calculations using the real ligands to validate the argument here (vide infra). Oxidative addition (OA) followed by concerted metalation deprotonation (CMD) gives the

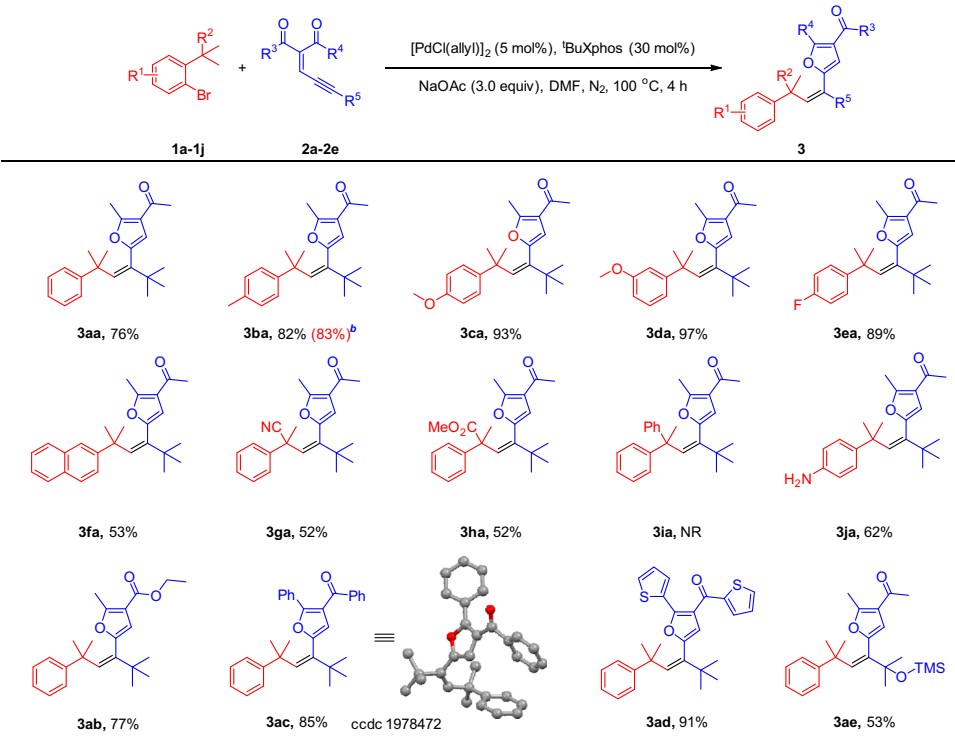

**Fig. 2 Substrate scopes of aryl bromides and ene-yne-ketones.** [a] Reaction conditions: **1** (0.1 mmol, 1.0 equiv), **2** (0.2 mmol, 2.0 equiv), [Pd(Cl(allyl)]₂ (5 mol%), ᵗBuXphos (30 mol%), NaOAc (3.0 equiv), DMF (1 mL), under N₂, 100 °C, 4 h. Yields of isolated products. [b] The reaction was run in 2.0 mmol scale.

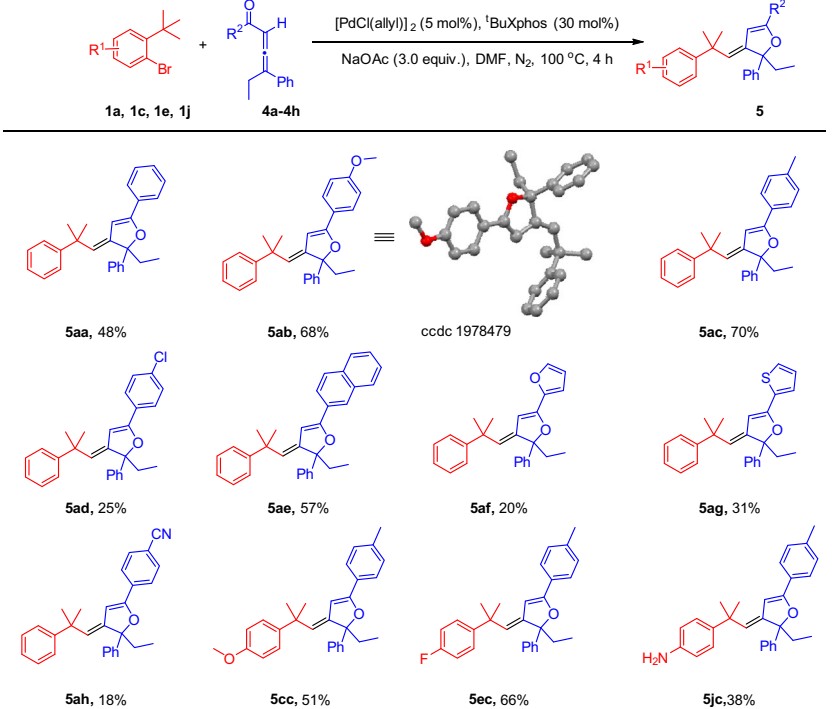

**Fig. 3 Substrate scopes of aryl bromides and allenes.** [a] Reaction conditions: **1** (0.1 mmol, 1.0 equiv), **4** (0.2 mmol, 2.0 equiv), [Pd(Cl(allyl)]₂ (5 mol%), ᵗBuXphos (30 mol%), NaOAc (3.0 equiv), DMF (1 mL), under N₂, 100 °C, 4 h. Yields of isolated products.

key palladacycle intermediate **IM4**. The barrier for the OA is small while the barrier for the CMD process is 25.0 kcal/mol.

From the key palladacycle intermediate **IM4**, two possible paths (consideration of the two cycles shown in Fig. 7) were

calculated (Fig. 8b). Path A involves alkyne-activation cyclization followed by migratory insertion (Cycle A in Fig. 7), while Path B engages protonation first and then alkyne-activation cyclization (Cycle B in Fig. 7). Clearly, Path A requires to pass through a very

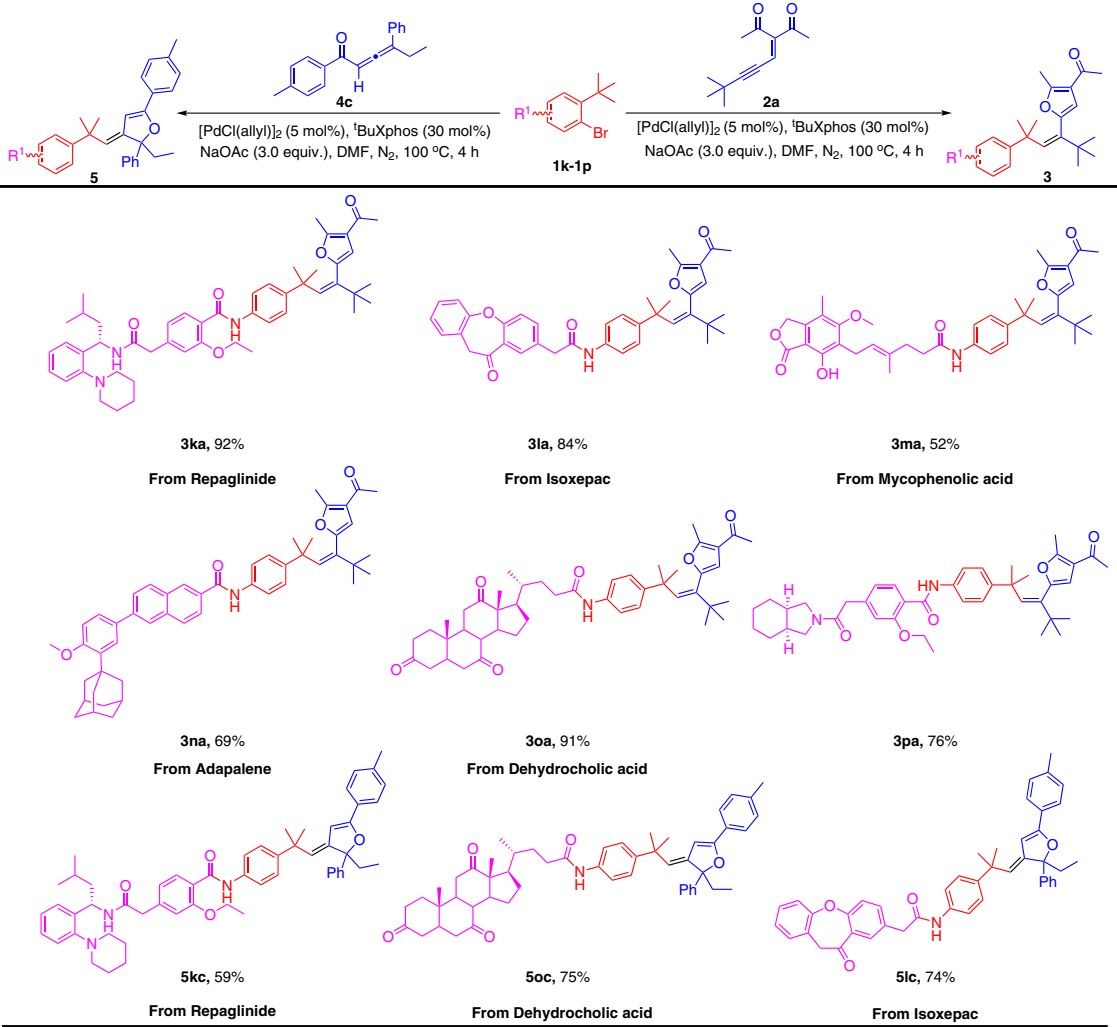

**Fig. 4 Substrate scopes of complex aryl bromides from modified drug molecules.** [a] Reaction conditions: **1k–1p** (0.1 mmol, 1.0 equiv), **2a** or **4c** (0.2 mmol, 2.0 equiv), [Pd(Cl(allyl)]₂ (5 mol%), ᵗBuXphos (30 mol%), NaOAc (3.0 equiv), DMF (1 mL), under N₂, 100 °C, 4 h. Yields of isolated products.

high-lying transition state (**TS₆₋₇ₐ**) for the migratory insertion. The high-lying **TS₆₋₇ₐ** structure is a result of the unfavorable migration step that involves weakening/breaking of the two strong Pd–C bonds in the five-membered ring of **IM6A**.

Figure 8c shows that when the migration insertion occurs on the carbene structure **IM7B** without a five-membered ring moiety, a very small barrier of 7.5 kcal/mol is calculated. After the migratory insertion, which is highly exergonic, β-hydride elimination occurs easily (almost barrierless), followed by reductive elimination and ligand (aryl bromide) coordination to regenerate the active species **A**. The β-hydride elimination in the agostic species **IM8B** gives the product **3a** having Z-selectivity. Here, one may wonder that **IM8B** undergoes a simple C–C bond rotation to give another isomeric agostic species from which a β-hydride elimination gives an E-product instead. Such an isomeric agostic species was calculated to be highly unstable (lying ca. 15 kcal/mol higher than **IM8B**; see Supplementary Fig 2) due to the excessive steric repulsion between the t-butyl and benzyl groups.

The calculation results suggest that Cycle B is favorable. From Fig. 8, we can also see that the CMD transition state structure **TS₂₋₃** (Fig. 8a) and the protonation transition state structure **TS₄₋₅ᵦ** (Path B in Fig. 8b) show similar stability, although the latter lies slightly higher than the former. On the basis of the results, **TS₄₋₅ᵦ** (Path B in Fig. 8b) is the rate-determining transition state, and therefore, the

overall reaction barrier is 25.8 kcal/mol, corresponding to the energy difference between **IM1** and **TS₄₋₅ᵦ**. In Fig. 8, the series of transformations from **IM2** to **IM5B** corresponds to a 1,4-Pd-shift.

Apart from all of the calculations mentioned above, we also calculated a pathway, which is closely related to Path A but starts from **IM4B** (instead of **IM4**) to react with **2a**. The calculation results (Supplementary Fig 1) indicate that this pathway is slightly favorable to Path A, but still less favorable than Path B.

Considering that the CMD process is energy-demanding (Fig. 8a), one may ask if it is possible for **IM1** to take in (coordinate with) the alkyne substrate molecule **2a** and undergo alkyne-activation cyclization (cf. Fig. 8b) to give a metal carbene intermediate (followed by Pd–carbene migration insertion) prior to the CMD process. Our calculations using the real ligand (Supplementary Fig. 3) indicate that such a process (alkyne-activation cyclization followed by Pd–carbene migration insertion) is indeed competitive with the CMD process. However, the C–H bond activation steps following such a process have inaccessibly high barriers. These results indicate that such a process is reversible, despite its competitive nature. In our proposed reaction mechanism, the alkyne-activation cyclization occurs in **IM5B** having a Pd–C(benzyl) bond, not in **IM1** or **IM2** having a Pd–C(aryl) bond. Our calculation results show that due to steric factor, coordination of the highly bulky substrate

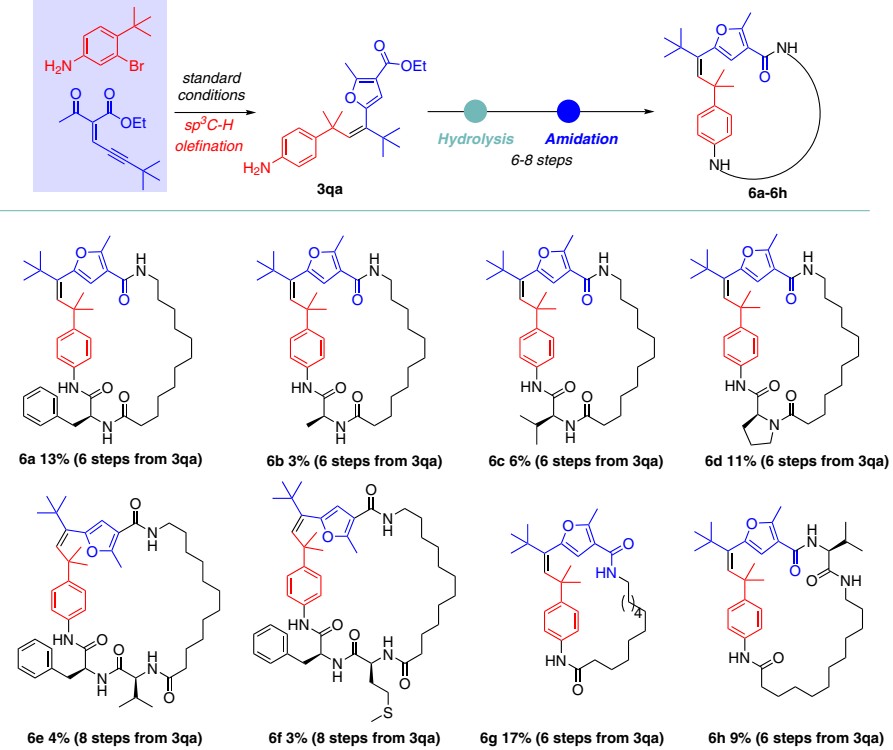

**Fig. 5 Construction of macrolactams using polysubstituted alkenes and natural or unnatural amino acids as building blocks.** Reaction conditions: please see page 7–14 of Supplementary Information.

molecule **2a** to a Pd(II) species having a Pd–C(aryl) bond is energetically much less favorable than to a Pd(II) species having Pd–C(benzyl) bond (Supplementary Figs. 3 and 4), explaining that CMD occurs in the currently reported coupling reaction with aryl bromides **1**, but not in a closely related coupling reaction with benzylic halides reported in the literature[49].

## Discussion

In summary, we develop two types of Pd-catalyzed intermolecular unactivated Csp³–H bond olefination mediated by choosing eneyne-ketones and allenyl ketones as donor/donor carbene precursors, allowing for the construction of a diversity of alkenes substituted with furans and dihydrofurans. These two carbene cross couplings exhibit high efficiency and stereoselectivity, which can be applied to late-stage cyclization/olefination of different therapeutic drugs. DFT mechanistic studies supported that a formal 1,4-Pd shift was involved in the catalytic cycle. Furthermore, alkenes substituted with furans as building blocks were successfully assembled via a short and modular biomimetic strategy into macrolactams, which showed significant anti-inflammatory activity with less cytotoxicity.

## Methods

**General information.** Nuclear magnetic resonance (NMR) spectra were recorded at room temperature on Bruker Avance III 400 Spectrometer (400 MHz) and Bruker Avance III 500 (Cryo) Spectrometer (500 MHz), using TMS as an internal standard. Chemical shifts are given in ppm and coupling constants in Hz. The following abbreviations were used for ¹H NMR spectra to indicate the signal multiplicities: s (singlet), d (doublet), t (triplet), q (quartet), and m (multiplet). High-resolution mass spectrometry was recorded on the Agilent G6520 Q-TOF. Chemicals were purchased from commercial suppliers. Unless stated otherwise, all the substrates and solvents were purified and dried according to standard methods prior to use.

**General procedure for unactivated Csp3–H olefination.** A screw-capped vial was charged with 2-bromo-1-(tert-butyl)-4-methylbenzene (0.1 mmol, 1.0 equiv.), the respective 3-(4,4-dimethylpent-2-yn-1-ylidene)pentane-2,4-dione (0.2 mmol,

2.0 equiv.), [Pd(Cl(allyl)]2 (5 mol%), tBuXphos (30 mol%), NaOAc (3.0 equiv.), DMF (1.0 mL). The reaction mixture was stirred at 100 °C under N₂ for 4 h, and then quenched with saturated aqueous NaCl and extracted with ethyl acetate. After drying over Na₂SO₄ for 30 min, the combined organic phase was concentrated, and the residue was purified by silica gel column chromatography with petroleum ether/ethyl acetate to afford the product as a yellow oil.

**Cell culture.** RAW264.7 cell, a murine macrophage cell line, was purchased from American Type Culture Collection (ATCC, Manassas, VA, USA). Cells were grown in Dulbecco's Modified Eagle Media (DMEM) containing 10% fetal bovine serum, 2 mM L-glutamine, 100 U/mL penicillin, and 100 μg/mL streptomycin and were maintained at 37 °C in a humidified incubator of 5% CO₂.

**Cell viability assay.** RAW264.7 cells (1 × 10⁵/well) were seeded into 96-well plates in triplicate and were treated with DMEM media or indicated concentrations of compounds for 24 h. MTT solution was added to each well at a final concentration of 0.5 mg/ml and incubated for 4 h before the end of the culture. The quantity of formazan is measured at 570 nm with a microplate reader (Molecular Devices, Sunnyvale, CA, USA) and the cell viability was calculated.

**Cytokines production assays.** RAW264.7 cells (1 × 10⁵/well) were seeded into 96-well plates in triplicate and were treated with DMEM media only or 5 μg/mL of LPS or with indicated concentrations of compounds for 24 h. Supernatants were collected and then quantified with the mouse TNF-α, IL-6, and IL-1β ELISA kits following the manufacturer's instructions.

**Western blotting.** RAW264.7 cells (6 × 10⁵/well) were seeded into 24-well plates and treated with DMEM media only or 5 μg/mL of LPS or with indicated concentrations of compounds for 24 h. Cells were lysed in sodium dodecyl sulfate (SDS) sample buffer containing protease inhibitor cocktail (Roche Life Science, Mannheim, Germany). The resulted proteins were resolved by SDS-polyacrylamide gel electrophoresis and then subjected to the following antibodies: anti-mouse IKK-α, anti-mouse phospho-IKK-α, anti-mouse IκB-α, anti-mouse NF-κB p65, anti-mouse phospho-NF-κB p65 antibodies (Cell Signaling Technology, Beverly, MA), and HRP-conjugated mouse anti-GAPDH (Kangcheng, Shanghai, China). Signals were detected with an HRP-conjugated anti-rabbit IgG (Bio-Rad, Richmond, CA, USA) using an ECL system (Amersham Biosciences, Buckinghamshire, UK). The uncropped and unprocessed scans of indicated blots were supplied as Supplementary Figure 99 in the Supplementary Information.

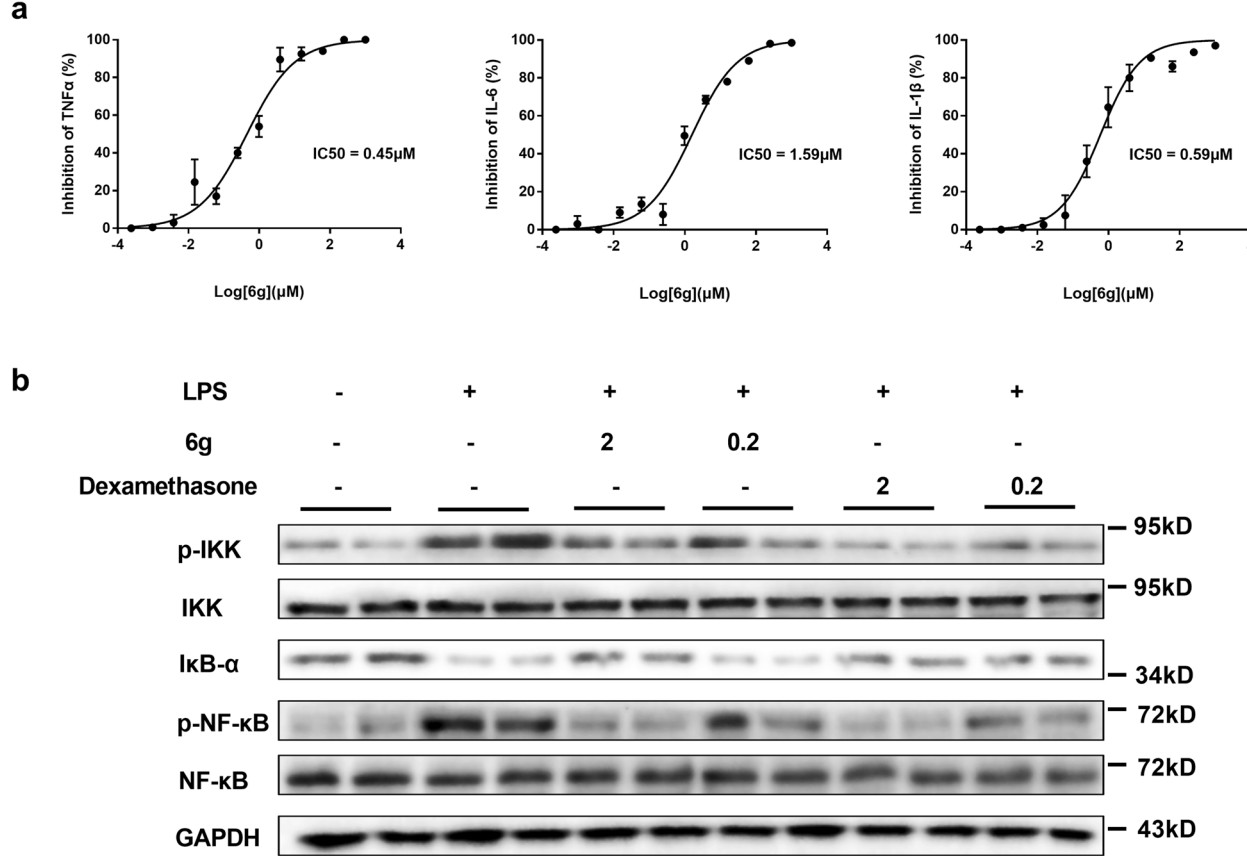

**Fig. 6 Compound 6g suppressed the LPS-induced inflammatory responses in RAW264.7 macrophages.** $IC_{50}$: the concentration of the compound needed to inhibit inflammatory mediators by 50% relative to the control value. **a** Effect of **6g** on LPS-induced TNF-α, IL-6, and IL-1β. Cells were treated with LPS alone (5 µg/mL) or with indicated concentrations of **6g** for 24 h. **b** Effect of **6g** on LPS-induced NF-κB activation. Dexamethasone was used as a positive control. $IC_{50}$ values were calculated by nonlinear regression (curve fit) applied log (inhibitor) vs. normalized response. Data are shown as the mean ± SEM, $n$ = biologically independent three wells for cytokine assays and two wells for Western blotting examined over two independent experiments.

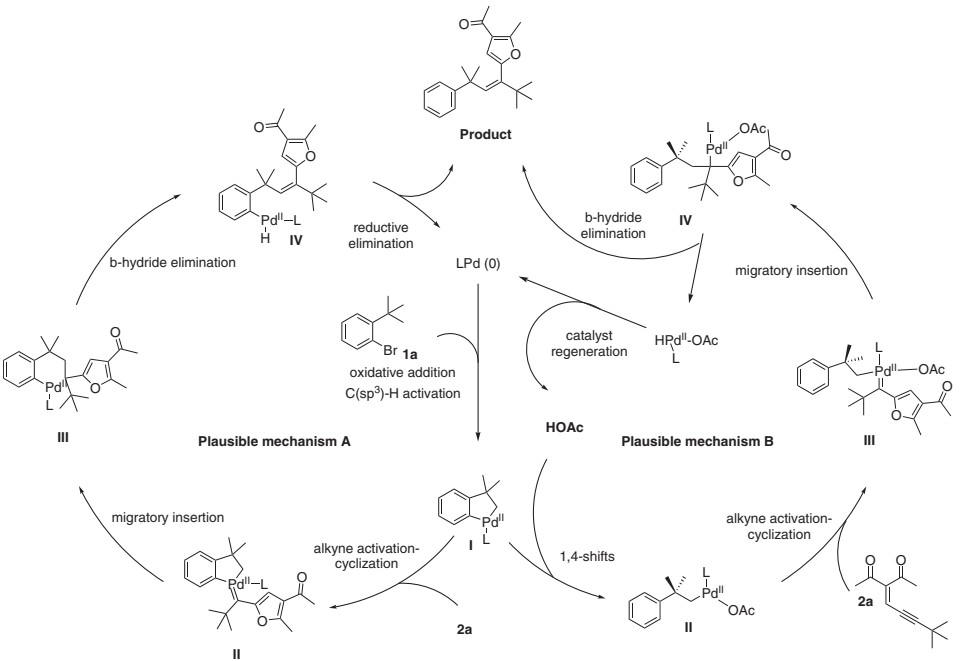

**Fig. 7 Two possible catalytic cycles for the reaction.** This proposed mechanism contains two possible paths, and path B is more favorable according to DFT calculation results.

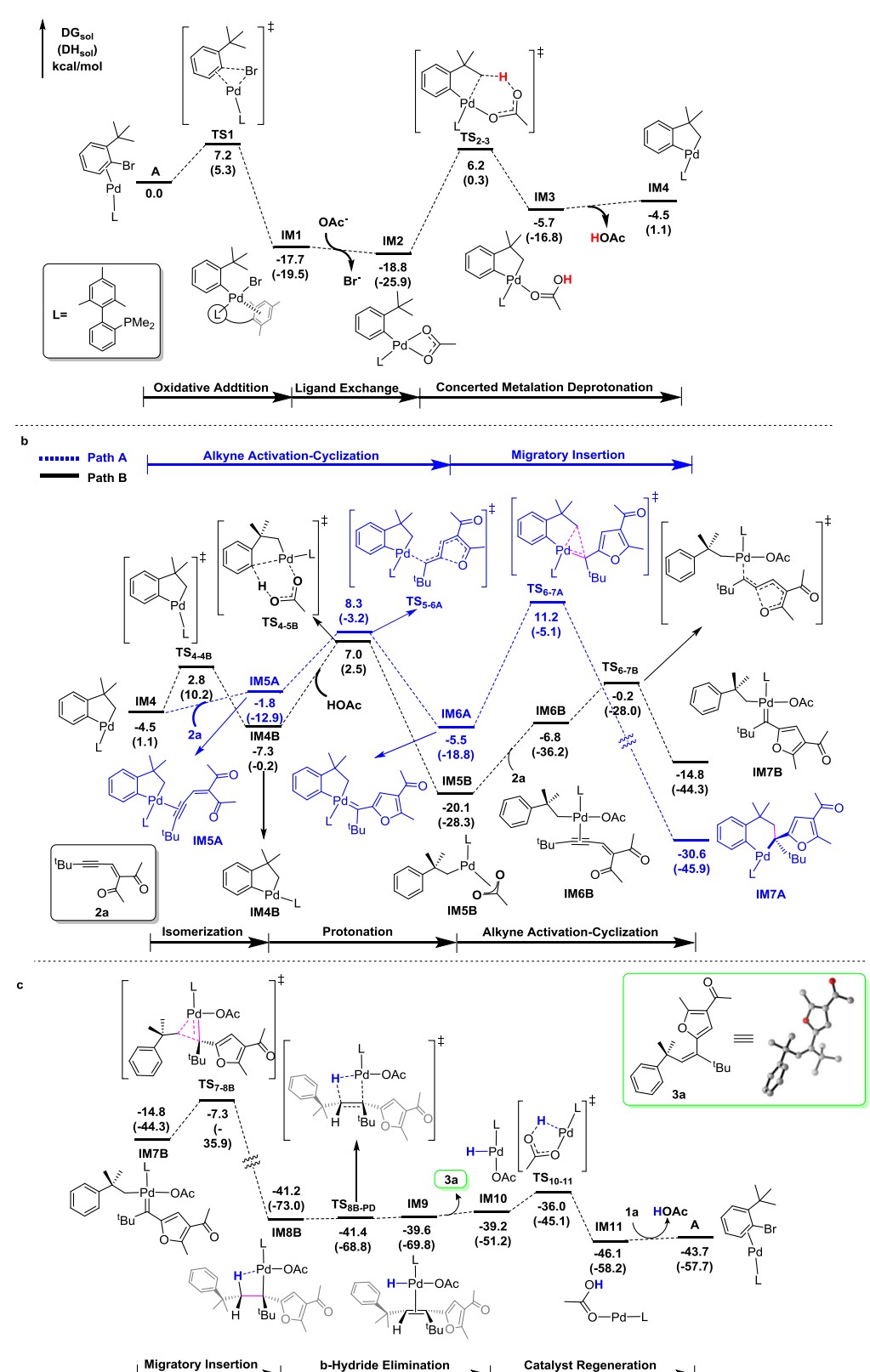

**Fig. 8 Energy profiles calculated on the basis of the mechanistic cycles shown in Fig. 7. a** Oxidative addition followed by CMD leading to the key palladacycle intermediate **IM4**, **b** two different paths starting from **IM4**, and **c** catalyst regeneration for the favorable path (Cycle B). The solvation-corrected relative free energies and electronic energies (in parentheses) are given in kcal/mol.

**Reporting summary**. Further information on research design is available in the Nature Research Reporting Summary linked to this article.

## Data availability

The X-ray crystallographic coordinates for compounds **3ac** and **5ab** have been deposited at the Cambridge Crystallographic Data Centre (CCDC), under deposition numbers 1978472 and 1978479. Other relevant data are available in Supplementary Information, Supplementary Data and from the authors.

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

## Acknowledgements
We gratefully acknowledge 100 talent program of Chinese Academy of Sciences, NSFC (21702217), "1000-Youth Talents Plan", Shanghai-Youth Talent, National Science & Technology Major Project" Key New Drug Creation and Manufacturing Program" China (No. 2018ZX09711002-006), Science and Technology Commission of Shanghai Municipality (18431907100) and Shanghai-Technology Innovation Action Plan (18JC1415300), and the Research Grants Council of Hong Kong (HKUST 16302418) for financial support of this research.

## Author contributions
Jiping Hao designed and carried out most of the chemical reactions and analyzed the data. Xueying Guo performed the DFT calculation. Shijun He and Jianping Zuo performed the biological experiments. Zhongliang Xu, Lu Chen, Zhongyu Li, and Bichao Song supported the performance of synthetic experiments and preparation of Supplementary Information. Weibo Yang and Zhenyang Lin designed the experiments. Weibo Yang conceived the idea and supervised the research. Weibo Yang and Zhenyang Lin prepared the paper.

## Competing interests
The authors declare no competing interests.
