## [Peer Review File · Nature Communications]

REVIEWER COMMENTS

Reviewer #1 (Remarks to the Author):

The manuscript from Lin and Yang have established an unactivated Csp³-H bond olefination methodology via donor/donor carbene migration insertion strategy from easily available ene-yne-ketone or allenyl ketone precursors. The integration of 1,4-Pd shift via C-H activation with well-known transition-metal-mediated enone cyclization of carbene precursors is very smart and elegant, which requires to balance the sequence for 1,4-Pd shift and carbene migrate insertion. In addition, the authors extended this methodology to furan-embedded macrolactam synthesis for bioactive screening, leading to discovery of 6g which has good activity in anti LPS-induced inflammation. These are impressive highlights of this manuscript. However, the authors have to revise their manuscript to fully address the following issues before a final decision is reached.

From the main text of this manuscript:

- a) Both the 1,4-Pd shift (Angew. Chem. Int. Ed. Engl. 1992, 31, 1023; Angew. Chem. Int. Ed. 2005, 44, 7512) and Pd-carbene migration insertion (J. Am. Chem. Soc. 2013, 135, 13502–13511; J. Org. Chem. 2019, 84, 8275–8283) are well-known process, the major advancement of this paper is to control the sequence for migration insertion. In my opinion, the DFT calculation didn't give a good explanation for the underlying principle of this chemistry. The authors should vary the tertbutyl group (t-Bu) of substrate 2: are there any possibility for the substrates bearing a secondary alkyl or aryl group at the alkyne terminal (instead the tertbutyl group in 2)? Whether a direct aryl migrate insertion process is dominated when aryl-capped 2 was used, owing to the very high activity barriers for CMD (33.4 kcal/mol)? Are the tertbutyl plays an important role in control both the Z/E geometry and retards the direct aryl migrate insertion before CMD occurring due to steric hindrance.
- b) Personally, this reaction reported herein is actually a Csp³-H functionalization reaction via a carbene migrate insertion (Chem. Rev. 2017, 117, 13810-13889), so the contents of the last part about the carbene insertion of C-H bond (Figure 1) are not suitable, which should be deleted. Similarly, the subtitle of "Pd-catalyzed intermolecular unactivated Csp³-H olefination via donor/donor carbenes insertion (Figure 1)" is a misused concept, which would be more reasonable for "Pd-catalyzed intermolecular unactivated Csp³-H olefination mediated by donor/donor carbenes". In addition, "metal-mediated cleavage of unactivated Csp³-H bonds is often significantly slower than Csp²-H bonds (in main text)" couldn't ascribe to or be equal to "low nucleophilic reactivity of Csp³-H bands (last part of Figure 1)". The greater activity for Csp²-H bonds might arise from the electrophilic nature of adjacent n-bond (Fridel-Crafts-like activation process), or a relative rigid conformation for concerted metalation-deprotonation (CMD) process. Actually, the carbene insertion process usually involved a fast hydride-transfer process (J. Am. Chem. Soc. 2000, 122, 3063-3070 and Angew. Chem. 2018, 130, 15433 –15436), while the Csp²-H bond couldn't act like that, which somehow demonstrated the good nucleophilic nature of Csp³-H bond in carbene insertion scenario. What's more, there are few cases of donor/donor carbene insertion of unactivated Csp³-H bond (Angew. Chem. 2018, 130, 15433–15436), which conflicts the author's statement of "no report". Last but not least, the alkyl migrate insertion of carbene [alkyl-Pd=C(donor)₂] is well-established, and the author overlook this background, which should be cited (J. Am. Chem. Soc. 2013, 135, 13502–13511 and J. Org. Chem. 2019, 84, 8275–8283). The author should focus on the difficulty of their reaction from thermodynamic and kinetic points, not on the irrelevant topic of carbene insertion of Csp³-H bond.
- c) Despite the author frequently mentioned the concept "modular biomimetic strategy" in this article, while, personally, the reviewer didn't get any sense of it from their synthesis routes or strategy, which actually a classical linear synthesis process. So, this statement need modification or deleted directly.
- d) As the author state: "A deeper examination into their scaffolds reflects that alkene substituted furan is of crucial importance for the biological activities", so some references should be cited.
- e) From the drug and lead compound design viewpoint, the huge structure hopping from the furanocembranoids to the macrolactams without previous research results or analysis, which were somewhat farfetched. In addition, there are no principle given for choosing the long-chain aminoacid and ring-size for lactam. So at least some analyses process should be given for the

readers to follow.

f) The olefination reaction in Table 4 is actually an olefination of drug derivatives by introducing a 1-bromo-2-(tert-butyl)benzene motif, not the drug itself (Table 4).

For SI:

- 1) New compounds should provide HRMS, such 1l, 1m, 1n, 1p;
- 2) The ¹H-NMR spectrum of 6c, 6d and 6f might contain undesired impurities or isomers, which need further purification. The intensity of ¹³C-NMR spectrum of 6c-6f is too low to identify, please improve it.

Reviewer #2 (Remarks to the Author):

Yang and co-workers describe the Pd-catalyzed C(sp³)-H insertion reaction involving ene-yne-ketones and allenyl ketones as carbene precursors which leads to the formation of furans and dihydrofurans. The scope of the process has been explored in both initial reactants and allows the access to a wide variety of (dihydro)furans. In addition, furan 3qa was applied to the construction of novel macrolactams whose biological activity has been also considered. Moreover, a computational (DFT) study of the possible mechanism involved in this transformation is also presented. Although this work is conceptually rather similar to the process reported by Martin and co-workers recently (JACS 2016, 138, 6384) and can be therefore considered as an extension of this previous work, I think that the authors have proven the synthetic utility of the process convincingly. For this reason, I support the publication of this work in a top journal such as Nat. Comm. Despite that, the following issues (mainly on the computational part of the work) should be addressed in a revision:

- (a) The authors confirmed the crucial role of the tBuXPhos in the process. However, a much simpler model of this ligand is used in the calculations which might severely affect the computed barriers. I therefore urge the authors to check the reliability of this change by calculating the barrier of the key steps with the real ligand.
- (b) According to Figure 3c, the computed beta-hydride elimination leads to the wrong alkene isomer (having the furan and H atom placed in relative cis-position). This originates from a wrong orientation in the coordination of the alkyne to the transition metal in IM6B. This should be corrected and both pathways (E/Z) should be compared and discussed in the manuscript to confirm the selectivity observed experimentally.
- (c) In my opinion, the term "1,4-Pd shift" is confusing, as one might expect a direct 1,4-migration of the transition metal fragment. According to the proposed mechanism, there is no such direct migration. So, if the authors still want to use this term, I would recommend to use "formal 1,4-Pd shift".
- (d) Figure 1 (pink block), "activitited" should be "activated". In addition, references should appear in previous works.

Reviewer #3 (Remarks to the Author):

On first sight this manuscript appears to describe a very novel site selective reaction. However, when one analyzes the reactions further, the C-H functionalization is actually not particularly novel at all. It relies on a site-selective reaction that was discovered a long time ago, I believe initially by Buchwald and has been extensively used by several groups, most notably Baudoin. Basically, when one has an aryl bromide with an ortho tert-butyl group, the sp² organopalladium will do a C-H functionalization at a methyl group of the tertbutyl to generate a sp³ organopalladium that can react further. This is what is happening in all of the examples in this paper.

The method trapping of the organopalladium is interesting but still well preceded. It is well established that the organometallic intermediate after C-H functionalization can react with carbene sources to ultimately lead to insertion into the carbene. In this paper the sp^3 organopalladium reacts with a donor donor carbene obtained from a cyclization reaction on alkynes. The resulting products look very nice and have been well selected, and the general way the paper has been presented is attractive. In my opinion, however, the novelty of the chemistry has been oversold and I am not able to support the publication of this work in Nature Communications in its current form. If it was rewritten and the work was put in the proper context, it would be worth considering further.

REVIEWER COMMENTS

Reviewer #1 (Remarks to the Author):

The manuscript from Lin and Yang have established an unactivated Csp³-H bond olefination methodology via donor/donor carbene migration insertion strategy from easily available ene-yne-ketone or allenyl ketone precursors. The integration of 1,4-Pd shift via C-H activation with well-known transition-metal-mediated enone cyclization of carbene precursors is very smart and elegant, which requires to balance the sequence for 1,4-Pd shift and carbene migrate insertion. In addition, the authors extended this methodology to furan-embedded macrolactam synthesis for bioactive screening, leading to discovery of 6g which has good activity in anti LPS-induced inflammation. These are impressive highlights of this manuscript. However, the authors have to revise their manuscript to fully address the following issues before a final decision is reached.

Response: Thank this reviewer for the positive comments on our manuscript.

From the main text of this manuscript:

a) Both the 1,4-Pd shift (Angew. Chem. Int. Ed. Engl. 1992, 31, 1023; Angew. Chem. Int. Ed. 2005, 44, 7512) and Pd-carbene migration insertion (J. Am. Chem. Soc. 2013, 135, 13502–13511; J. Org. Chem. 2019, 84, 8275–8283) are well-known process, the major advancement of this paper is to control the sequence for migration insertion. In my opinion, the DFT calculation didn't give a good explanation for the underlying principle of this chemistry. The authors should vary the tertbutyl group (t-Bu) of substrate 2: are there any possibility for the substrates bearing a secondary alkyl or aryl group at the alkyne terminal (instead the tertbutyl group in 2)? Whether a direct aryl migrate insertion process is dominated when aryl-capped 2 was used, owing to the very high activity barriers for CMD (33.4 kcal/mol)? Are the tertbutyl plays an important role in control both the Z/E geometry and retards the direct aryl migrate insertion before CMD occurring due to steric hindrance.

Response: Thank the reviewer for the insightful comments. We tested the substrates bearing a secondary alkyl or aryl group at the alkyne terminal, however, they exhibited no activity under the standard conditions and only the starting materials were recovered. Moreover, we have carried out additional calculations for the possible pathway related to alkyne-activation cyclization followed by Pd-carbene migration insertion before the energy-demanding CMD process. To have a more accurate comparison, we used the experimentally employed ligand in our calculations. The results from the additional calculations indicate that the possible pathway is reversible and competes against the CMD process. The reversibility is due to that the corresponding CMD steps following the Pd-carbene migration insertion are kinetically very unfavorable as a result of 4- and 6-membered-ring transition states. In our proposed reaction mechanism, the alkyne activation cyclization occurs in **IM5B** having a Pd-C(benzyl) bond, not in **IM1** or **IM2** having a Pd-C(aryl) bond. Due to steric factor,

coordination of the highly bulky substrate molecule **2a** to a Pd(II) species having a Pd-C(aryl) bond is energetically much less favorable than to a Pd(II) species having Pd-C(benzyl) bond, explaining that CMD occurs in the coupling reaction with aryl bromides **IM1**, but not in a closely related coupling reaction with benzylic halides reported in the literature (*J. Am. Chem. Soc.* 2013, 135, 36, 13502–13511. This reference is now cited).

Figure 1. Energy profiles calculated (using the experimentally employed ligand) for an alternative pathway considering an alkyne activation cyclization immediately after oxidative addition. The results here indicate that such an alkyne activation cyclization is reversible in view of the inaccessibly high lying **TS_{5-6-b-r-1}** and **TS_{5-6-b-r-2}**. The solvation-corrected relative free energies and electronic energies (in parentheses) are given in kcal/mol.

^a through 6-membered ring

^b through 4-membered ring

Figure 2. Energy profile calculated for alkyne activation cyclization using the experimentally employed ligand. The barrier is compatible with that obtained by using the model ligand (cf. Figure 3(b) in the main text), but is much smaller than the corresponding step shown in Figure S3 (from **IM1-r** to **TS_{2-3-b-r}**). The solvation-corrected relative free energies and electronic energies (in parentheses) are given in kcal/mol.

For the *Z/E* selectivity, we also added calculations for the pathway leading to an *E*-product. The results show that the excessive steric repulsion between the *t*-butyl and benzyl groups hinders the generation of the *E*-product.

Brief discussion similar to what we described here has been added in the main text, and corresponding energy profiles are put in the supporting information.

Figure 3. Energy profile calculated for the β -Hydride elimination from a highly unstable isomeric agostic species (cf. IM8B in Figure 3(c) in the main text). The solvation-corrected relative free energies and electronic energies (in parentheses) are given in kcal/mol.

b) Personally, this reaction reported herein is actually a Csp³-H functionalization reaction via a carbene migrate insertion (Chem. Rev. 2017, 117, 13810-13889), so the contents of the last part about the carbene insertion of C-H bond (Figure 1) are not suitable, which should be deleted. Similarly, the subtitle of "Pd-catalyzed intermolecular unactivated Csp³-H olefination via donor/donor carbenes insertion (Figure 1)" is a misused concept, which would be more reasonable for "Pd-catalyzed intermolecular unactivated Csp³-H olefination mediated by donor/donor carbenes". In addition, "metal-mediated cleavage of unactivated Csp³-H bonds is often significantly slower than Csp²-H bonds (in main text)" couldn't ascribe to or be equal to "low nucleophilic reactivity of Csp³-H bands (last part of Figure 1)". The greater activity for Csp²-H bonds might arise from the electrophilic nature of adjacent π -bond (Friedel-Crafts-like activation process), or a relative rigid conformation for concerted metalation-deprotonation (CMD) process. Actually, the carbene insertion process usually involved a fast hydride-transfer process (J. Am. Chem. Soc. 2000, 122, 3063-3070 and Angew. Chem. 2018, 130, 15433–15436), while the Csp²-H bond couldn't act like that, which somehow demonstrated the good nucleophilic nature of Csp³-H bond in carbene insertion scenario. What's more, there are few cases of donor/donor carbene insertion of unactivated Csp³-H bond (Angew. Chem. 2018, 130, 15433–15436), which conflicts the author's statement of "no report". Last but not least, the alkyl migrate insertion of carbene [alkyl-Pd=C(donor)₂] is well-established, and the author overlook this background, which should be cited (J. Am. Chem. Soc. 2013, 135, 13502–13511 and J. Org. Chem. 2019, 84, 8275–8283). The author should focus on the difficulty of their reaction from thermodynamic and kinetic points, not on the irrelevant topic of carbene insertion of Csp³-H bond.

Response: Thank the reviewer for the insightful comments. To address the concerns and criticism, we have deleted the unsuitable description and have rewritten the Introduction section. Moreover, we have added the related references.

c) Despite the author frequently mentioned the concept "modular biomimetic strategy" in this article, while, personally, the reviewer didn't get any sense of it from their synthesis routes or strategy, which actually a classical linear synthesis process. So, this statement need modification or deleted directly.

Response: We are very sorry to bring the concern to this reviewer. A biomimetic strategy could be described by two parts: mimicking a biosynthetic pathway or using biomimetic building blocks. In our case, it is a modular biomimetic strategy, which simply uses the fundamental building blocks from living organism's endogenous ligand, such as amino acids. Consequently, this strategy could enhance the hit rate of bioactive compounds. Employing modular biomimetic synthesis strategy is our key design concept, please also see our recent publication (W. Yang et. al., Nat. Commun. 2020,

11, 2151.) and we hope we can keep this concept in the manuscript.

d) As the author state: "A deeper examination into their scaffolds reflects that alkene substituted furan is of crucial importance for the biological activities", so some references should be cited.

Response: Thanks for this reviewer's suggestion. We revised this statement and have added some related references in the updated manuscript.

e) From the drug and lead compound design viewpoint, the huge structure hopping from the furanocembranoids to the macrolactams without previous research results or analysis, which were somewhat farfetched. In addition, there are no principle given for choosing the long-chain amino acid and ring-size for lactam. So at least some analyses process should be given for the readers to follow.

Response: Thank the reviewer for the insightful comments. On one hand, these long-chain amino acid are simple enough in structure, convenient to be obtained, and ready for cyclization reaction. On the other hand, very recently, we have successfully assembled the polysubstituted butadienes and long-chain amino acid into the macrolactams, which efficiently inhibited the P-glycoprotein and dramatically reversed multidrug resistance in cancer cells by 190-fold (W. Yang et. al., J. Am. Chem. Soc. 2020, 142, 9982-9992). Although the exact reason was still unclear at the moment, we speculated that these long-chain amino acid could change the conformation of macrolactams and make the ring not rigid. Based on this discovery, we wanted to test the generality of long-chain amino acid in other macrocycles.

f) The olefination reaction in Table 4 is actually an olefination of drug derivatives by introducing a 1-bromo-2-(tert-butyl)benzene motif, not the drug itself (Table 4).

Response: Thank the reviewer for the insightful comments. We have revised the related text in the updated manuscript.

For SI:

1) New compounds should provide HRMS, such 1l, 1m, 1n, 1p;

2) The ¹H-NMR spectrum of 6c, 6d and 6f might containi undesired impurities or isomers, which need further purification. The intensity of ¹³C-NMR spectrum of 6c-6f is too low to identify, please improve it.

Response: Thank the reviewer for the insightful comments. The purification and characterizations of the mentioned compounds have been done and improved. Please see the revised SI.

Reviewer #2 (Remarks to the Author):

Yang and co-workers describe the Pd-catalyzed C(sp³)-H insertion reaction involving enyne-ketones and allenyl ketones as carbene precursors which leads to the formation of furans and dihydrofurans. The scope of the process has been explored in both initial reactants and allows the access to a wide variety of (dihydro)furans. In addition, furan 3qa was applied to the construction of novel macrolactams whose biological activity has been also considered. Moreover, a computational (DFT) study of the possible mechanism involved in this transformation is also presented. Although this work is conceptually rather similar to the process reported by Martin and co-workers recently (JACS 2016, 138, 6384) and can be therefore considered as an extension of this previous work, I think that the authors have proven the synthetic utility of the process convincingly. For this reason, I support the publication of this work in a top journal such as Nat. Comm. Despite that, the following issues

(mainly on the computational part of the work) should be addressed in a revision:

Response: Thank the reviewer for the positive comments and supporting publication of this work.

(a) The authors confirmed the crucial role of the tBuXPhos in the process. However, a much simpler model of this ligand is used in the calculations which might severely affect the computed barriers. I therefore urge the authors to check the reliability of this change by calculating the barrier of the key steps with the real ligand.

Response: Thank the reviewer for the insightful comments. As suggested, we have calculated several key steps by using the real (experimentally employed) ligand. The results show that the total barrier is raised up by 3.9 kcal/mol, but still reasonable considering the reaction temperature is 100°C. And the trend is not changed as what we have calculated by using the model ligand.

(b) According to Figure 3c, the computed beta-hydride elimination leads to the wrong alkene isomer (having the furan and H atom placed in relative cis-position). This originates from a wrong orientation in the coordination of the alkyne to the transition metal in IM6B. This should be corrected and both pathways (E/Z) should be compared and discussed in the manuscript to confirm the selectivity observed experimentally.

Response: Thank the reviewer for pointing out this mistake. The IM6B was wrongly drawn. Now the mistake has been corrected. For the discussion of E/Z selectivity, reviewer 1 also pointed out this. We have addressed this by carrying out additional calculations. Please see our response above.

(c) In my opinion, the term "1,4-Pd shift" is confusing, as one might expect a direct 1,4-migration of the transition metal fragment. According to the proposed mechanism, there is no such direct migration. So, if the authors still want to use this term, I would recommend to use "formal 1,4-Pd shift".

Response: Thanks for this reviewer's suggestion. We have changed "1,4-Pd shift" and used "formal 1,4-Pd shift" in the revised manuscript.

(d) Figure 1 (pink block), "activitited" should be "activated". In addition, references should appear in previous works.

Response: Sorry for this mistake. We have changed "activitited" into "activated" and added references in Figure 1.

Reviewer #3 (Remarks to the Author):

On first sight this manuscript appears to describe a very novel site selective reaction. However, when one analyzes the reactions further, the C-H functionalization is actually not particularly novel at all. It relies on a site-selective reaction that was discovered a long time ago, I believe initially by Buchwald and has been extensively used by several groups, most notably Baudoin. Basically, when one has an aryl bromide with an ortho tert-butyl group, the sp^2 organopalladium will do a C-H functionalization at a methyl group of the tertbutyl to generate a sp^3 organopalladium that can the react further. This is what is happening in all of the example in this paper.

The method trapping of the organopalladium is interesting but still well precedented. It is well established that the organometallic intermediate after C-H functionalization can react with carbene sources to ultimately lead to insertion into the carbene. In this paper the sp^3 organopalladium reacts with a donor donor carbene obtained from a cyclization reaction on alkynes. The resulting products look very nice and have been well selected, and the general way the paper has been presented is attractive. In my opinion, however, the novelty of the chemistry has been oversold and I am not able to support the publication of this work in Nature Communications in its current form. If it was rewritten and the work was put in the proper context, it would be worth considering further.

Response: Thanks for this reviewer's suggestion, we have rewritten the Introduction section and added previously published work about trapping transformations of σ -alkylpalladium via C(sp^3)-H bond activation/formal 1,4-Pd shift process in the revised manuscript.

REVIEWER COMMENTS

Reviewer #1 (Remarks to the Author):

I have carefully checked the revised manuscript and the supporting informations. All the issues raised by the reviewers have been well addressed. It can be accepted as it is.

Reviewer #2 (Remarks to the Author):

The authors have responded to the points I originally made and I am happy to recommend acceptance of this fine manuscript as it stands.

Reviewer #3 (Remarks to the Author):

Even though there are elements of novelty associated with this paper, the authors are caught up in fancy terminology that do not accurately represent their work. One area that I mentioned before was that their chemistry does not represent an intermolecular C-H functionalization with donor carbenes – it is an intramolecular C-H activation followed by reaction of the alkyl palladium complex with the carbene to ultimately lead to the coupled product. There are examples of such intermolecular reactions with acceptor and donor/acceptor carbenes where those carbenes do directly react intermolecularly with unactivated C-H bonds (which they reference) but their work is not a similar reaction with donor carbenes. Hence they need to make this clear. The current statement shown below need to be adjusted accordingly.

“First, to the best of our knowledge, although advances have been made in acceptor carbene cross-couplings of activated Csp³ -H bonds²⁷⁻³² or even unactivated Csp³ -H bonds³³⁻³⁹, or intermolecular carbene cross-couplings of unactivated Csp³ -H bonds with donor/donor carbene precursors have never been documented. It is noteworthy that the electrophilic capacity of acceptor carbene is relatively strong compared to the donor/donor carbene. Second, while a few donor/donor carbene cross-couplings of Csp³ -H substrates have been developed, they are mainly limited to activated positions (benzylic, allylic, α-to heteroatoms)⁴⁰, or restricted to intramolecular version⁴¹⁻⁴² .” It is important to put the work in proper context and it is still not there.

I also agree with one of the other reviewers that the term “biomimetic modularization” and “biomimetic strategy” makes no sense in the context of this paper and should be removed.

REVIEWER COMMENTS

Reviewer #1 (Remarks to the Author):

I have carefully checked the revised manuscript and the supporting informations. All the issues raised by the reviewers have been well addressed. It can be accepted as it is.

Response: Thank the reviewer for the positive comments and supporting publication of this work.

Reviewer #2 (Remarks to the Author):

The authors have responded to the points I originally made and I am happy to recommend acceptance of this fine manuscript as it stands.

Response: Thank the reviewer for the positive comments and supporting publication of this work.

Reviewer #3 (Remarks to the Author):

a) Even though there are elements of novelty associated with this paper, the authors are caught up in fancy terminology that do not accurately represent their work. One area that I mentioned before was that their chemistry does not represent an intermolecular C-H functionalization with donor carbenes - it is an intramolecular C-H activation followed by reaction of the alkyl palladium complex with the carbene to ultimately lead to the coupled product. There are examples of such intermolecular reactions with acceptor and donor/acceptor carbenes where those carbenes do directly react intermolecularly with unactivated C-H bonds (which they reference) but their work is not a similar reaction with donor carbenes. Hence they need to make this clear. The current statement shown below need to be adjusted accordingly. "First, to the best of our knowledge, although advances have been made in acceptor carbene cross-couplings of activated Csp³-H bonds²⁷⁻³² or even unactivated Csp³-H bonds³³⁻³⁹, or intermolecular carbene cross-couplings of unactivated Csp³-H bonds with donor/donor carbene precursors have never been documented. It is noteworthy that the electrophilic capacity of acceptor carbene is relatively strong compared to the donor/donor carbene. Second, while a few donor/donor carbene cross-couplings of Csp³-H substrates have been developed, they are mainly limited to activated positions (benzylic, allylic, α -to heteroatoms)⁴⁰, or restricted to intramolecular version⁴¹⁻⁴² ." It is important to put the work in proper context and it is still not there.

Response: Thank the reviewer for the insightful comments. Herein we want to make a further statement about the difference between the referenced works and our current

work. First, Csp^3-H carbene couplings have two distinct pathways: (I) metal carbene complex first formed and subsequent Csp^3-H bond activation/migratory insertion occurred. (II) Csp^3-H bond activation initially occurred followed by carbene formation/migratory insertion. Compared to path II, path I has been widely investigated. Our case belongs to the path II. We had highlighted the revised part with gray color. For more details, please see the revised manuscript.

b) I also agree with one of the other reviewers that the term “biomimetic modularization” and “biomimetic strategy” makes no sense in the context of this paper and should be removed.

Response: We are very sorry to bring the concern to this reviewer. Yes, at the very beginning, the reviewer 1# also had this concern on biomimetic strategy. However, we made a further explanation to the reviewer 1#. Finally, it seems that the reviewer 1# is happy about our statement. Right now, we want to explain it again to this reviewer. A biomimetic strategy could be described by two parts: mimicking a biosynthetic pathway or using biomimetic building blocks. In our case, it is a modular biomimetic strategy, which simply uses the fundamental building blocks from living organism's endogenous ligand, such as amino acids. Consequently, this strategy could enhance the hit rate of bioactive compounds. Employing modular biomimetic synthesis strategy is our key design concept, please also see our recent publication (W. Yang et. al., Nat. Commun. 2020, 11, 2151.) and we hope we can keep this concept in the manuscript.

REVIEWERS' COMMENTS

Reviewer #3 (Remarks to the Author):

The authors have adequately addressed in the written portion the difference between their approach for C-H functionalization and a direct carbene induced intermolecular C-H functionalization. Still casual reader glancing at figure 1 in the design section in blue would think of an intermolecular C-H functionalization whereas the reaction is actually an intramolecular C-H activation followed by reaction of the alkyl metal complex with the carbene. However, I am generally OK with this because if one reads carefully one would understand what is happening.

I still consider describing the chemistry as a biomimetic strategy as inappropriate. It sounds nice but I still do not see this work as biomimetic.

REVIEWER COMMENTS

Reviewer #3 (Remarks to the Author):

The authors have adequately addressed in the written portion the difference between their approach for C-H functionalization and a direct carbene induced intermolecular C-H functionalization. Still casual reader glancing at figure 1 in the design section in blue would think of an intermolecular C-H functionalization whereas the reaction is actually an intramolecular C-H activation followed by reaction of the alkyl metal complex with the carbene. However, I am generally OK with this because if one reads carefully one would understand what is happening.

I still consider describing the chemistry as a biomimetic strategy as inappropriate. It sounds nice but I still do not see this work as biomimetic.

Response: Thank this reviewer for the positive comments on our manuscript. Employing modular biomimetic synthesis strategy is our key design concept, please also see our recent publication (W. Yang et. al., Nat. Commun. 2020, 11, 2151.) and we hope we can keep this concept in the manuscript.